# Anatomical Characteristics Predict Response to Transcranial Direct Current Stimulation (tDCS): Development of a Computational Pipeline for Optimizing tDCS Protocols

**DOI:** 10.3390/bioengineering12060656

**Published:** 2025-06-15

**Authors:** Giulia Caiani, Emma Chiaramello, Marta Parazzini, Eleonora Arrigoni, Leonor J. Romero Lauro, Alberto Pisoni, Serena Fiocchi

**Affiliations:** 1Dipartimento di Elettronica, Informazione e Bioingegneria (DEIB), Politecnico di Milano, 20133 Milan, Italy; 2Institute of Electronics, Computer and Telecommunication Engineering (IEIIT), National Research Council (CNR), 20133 Milan, Italy; emma.chiaramello@cnr.it (E.C.); marta.parazzini@cnr.it (M.P.); serena.fiocchi@cnr.it (S.F.); 3Department of Psychology, University of Milano–Bicocca, 20126 Milan, Italy; eleonora.arrigoni@unimib.it (E.A.); leonor.romero1@unimib.it (L.J.R.L.); alberto.pisoni@unimib.it (A.P.)

**Keywords:** anodal tDCS, clinical outcome, computational modelling, cortical excitability, neuromodulation, personalized medicine

## Abstract

Transcranial direct current stimulation (tDCS) is a non-invasive brain stimulation technique promisingly used to treat neurological and psychological disorders. Nevertheless, the inter-subject heterogeneity in its after-effects frequently limits its efficacy. This can be attributed to fixed-dose methods, which do not consider inter-subject anatomical variations. This work attempts to overcome this constraint by examining the effects of age and anatomical features, including the volume of cerebrospinal fluid (CSF), the thickness of the skull, and the composition of brain tissue, on electric field distribution and cortical excitability. A computational approach was used to map the electric field distribution over the brain tissues of realistic head models reconstructed from MRI images of twenty-three subjects, including adults and children of both genders. Significant negative correlations (*p* < 0.05) were found in the data between the maximum electric field strength and anatomical variable parameters. Furthermore, this study showed that the percentage of brain tissue exposed to an electric field amplitude above a pre-defined threshold (i.e., 0.227 V/m) was the main factor influencing the responsiveness to tDCS. In the end, the research suggests multiple regression models as useful tool to predict subjects’ responsiveness and to support a personalized approach that tailors the injected current to the morphology of the patient.

## 1. Introduction

In recent years, non-invasive brain stimulation (NIBS) techniques have been explored as tools to restore proper functionality in the altered brain networks that characterize neuropsychological and psychiatric disorders [1,2,3,4]. These techniques induce neuroplastic changes that can rebalance the maladaptive activity and functional connectivity between brain structures [5].

Among various NIBS techniques, transcranial direct current stimulation (tDCS) is one of the most promising and accessible because of its low cost, ease of use, and tolerability. It is based on the delivery of a weak constant current, typically ranging from 0.5 to 4 mA, through electrodes applied over the scalp [6,7,8,9]. In anodal tDCS the current flows from the anode (positive electrodes) to the cathode (negative electrodes), and the stimulation is generated by a battery-driven constant current device to ensure safety and stability. The stimulation usually lasts from 10 to 30 min and includes short fade-in and fade-out periods to gradually ramp the current up and down, minimizing potential skin sensations.

tDCS modulates neural activity by interacting with the brain’s endogenous neuronal processes through the application of a weak electrical current. Computational modelling studies have shown that, owing to the low current intensity used in tDCS protocols, the resulting electric field in brain tissues is typically around hundreds of mV/m [10,11]. The induced electric field (EF), although too week to directly trigger action potentials, can still exert a significant neuromodulatory effect. This occurs through subthreshold shifts in the membrane potential of neurons, which may lead to either depolarization or hyperpolarization depending on the orientation of neurons relative to the direction of the electric field [12,13,14].

It has been demonstrated that the increase in cortical excitability induced by tDCS persists beyond the stimulation period and can induce plastic changes resembling long-term potentiation (LTP) or long-term depression (LTD) [15].

tDCS’s efficacy has been investigated for different neuropsychiatric and neurological disorders (for a detailed review of tDCS applications see, e.g., [16]) including Parkinson’s [17] and Alzheimer’s diseases [18], schizophrenia, and anxiety disorder. Recent meta-analyses and systematic reviews have demonstrated the efficacy of tDCS in significantly enhancing performance on cognitive tasks [19]. Regarding depression, Razza and colleagues [20] reported that tDCS is modestly effective in treating depressive episodes, while Moffa et al. [21] found a significant improvement in major depression disorder symptoms following tDCS sessions.

One of the most used methods for the assessment of tDCS-induced neurophysiological outcomes is based on the measurement of motor evoked potentials (MEPs), elicited through the application of Transcranial Magnetic Stimulation (TMS) pulses over the motor cortex [22]. However, when looking at responses outside the primary motor or sensory cortices, the most efficient neurophysiological index is TMS evoked potentials (TEPs), which are obtained through the integration of TMS and Electroencephalography (TMS-EEG) and allow one to explore the state of momentary excitability of the brain. TEPs, particularly their early components, have been demonstrated to be valuable measures of the cortical excitability of the brain in several studies [5,23,24,25,26,27].

This study focused on the application of anodal tDCS to the right posterior parietal cortex (PPC). The PPC plays a key role in different sensorimotor and cognitive functions [28,29], specifically in visuospatial attention, a function often compromised in various neurological and psychiatric conditions [30,31,32,33,34,35,36]. Several studies have shown that tDCS can be successfully applied to this area to modulate sensory and cognitive processing in healthy participants. In particular, research has shown that anodal stimulation of the rPPC can enhance the performance in tasks requiring visuospatial attention, such as visual search and spatial orientation [28,31,33], and improve neglect symptoms, usually found in individuals with stroke or other neurological impairments [36]. Despite its growing application in clinical settings, tDCS’s use is still scarce due to its low reproducibility, high variability, and poor experimental design [37].

So far, indeed, tDCS protocols are based on a “one-size-fits-all” approach that does not consider inter-subject variability in terms of anthropometric quantities, causing a significant portion of non-respondent subjects [38].

Computational models [39,40,41,42,43] have suggested that this is due to the different spatial distributions of the induced electric field across subjects, despite the same applied protocol in terms of electrode montage and delivered stimulation dose [44]. Consequently, the “one-size-fits-all” approach must be replaced by more personalized strategies to maximize the responsiveness of the subjects. Relying on uniform protocols limits the effectiveness and clinical applicability of tDCS, as it fails to account for inter-individual variability in anatomy and treatment response.

The EF distribution, which is the main determinant of the physiological response, is highly variable due to a combination of characteristics of the input electric signal (i.e., the stimulation device parameters) and individual tissue geometries and electrical properties [45]. Specifically, anthropometric characteristics, such as skull thickness, cortex folding, and cerebrospinal fluid (CSF) volume, significantly alter how current flows through brain tissues, resulting in a wide range of EF distributions [10,43,46,47,48,49].

Among the anatomical characteristics that can be effectively included in optimization protocols, skull thickness is defined as the distance between the skull’s outer surface and the CSF–bone interface, which is a crucial determinant of current’s passage from the skin into the brain [50]; the gyral and sulcal anatomy (cortex folding) influence how the electric field stimulates different brain regions [51,52,53], whereas CSF shunts the current away from the targeted region, due to its much higher conductivity with respect to the other tissues [43,54]. Furthermore, state-based factors, such as fatigue, time of day, and activities prior to or during stimulation, also play a role in shaping the EF distribution [55,56].

In this context, the aim of this work is to investigate how anatomical characteristics influence the electric field distribution induced during tDCS protocols and how they influence, in turn, subjects’ responsiveness to the stimulation. This aim was achieved through a computational approach, by combining realistic subject-specific models, obtained by MRI scans, with advanced solvers for electromagnetic problems. This approach provides a deeper understanding of the spatial distribution of EF in brain tissues and allows us to address the correlations between anatomical differences and EF distribution, improving the understanding of how these variables affect tDCS outcomes. The employment of subject-specific models allows the identification of the critical features most affecting tDCS outcomes, providing a unique insight into how to shift from a fixed-dose approach to a personalized stimulation protocol.

## 2. Materials and Methods

### 2.1. Databases–Anatomical Modelling

Simulations were performed on two different head models’ databases as follows:

*Clinical Outcome Database (CODb)*—It is composed of twelve Caucasian subjects (aged between 22–38) that took part in the study by Romero and colleagues [24], in which clinical outcomes, TEPs, were recorded (for methodological details, refer to Romero et al. [24]). The head models were reconstructed in a voxel-based format by the segmentation of high-resolution (i.e., 1 mm) T1-weighted structural MRIs. The segmentation was performed using the Sim4Life V8.2 [57] eHead40 function, which allows one to distinguish up to forty different tissues—including skin, CSF, bone cortical, bone cancellous, brain grey and white matter and internal air—and to precisely determine the position of the electrodes on the scalp according to the EEG 10-10 system [58]. In this way consistency of the position of the electrodes holds across simulations.

*Integrated Database (IDb)*—It includes highly detailed anatomical models. It contains eight models belonging to the “Virtual Family” [59]: two adults Ella and Duke, 26- and 34-year-olds, respectively; two adolescents Billie and Louis, 11- and 14-year-olds, respectively; and four child models, aged between 5 and 8 years old. All are segmented in a voxel-based format at a resolution of less than 1 mm, and they include up to 84 different tissues and organs over the whole-body. The database also includes a MIDA head model, a high-resolution multimodal imaging-based anatomical model of a 29 year old [60], in which a total of 153 different regions are distinguished only in the area of the head; the other two subjects, both 40 years old, were reconstructed from two 3T-MRIs of healthy volunteers provided by the same research group of the Romero and colleagues study, following the same reconstruction approach used for *CODb* subjects [24] (in the following, named “Head Reconstruction–HR1” and “HR2”).

The *IDb* was further distinguished in subjects over and under 20 years old to allow a direct and coherent comparison with the *CODb*. A summary of the anatomical quantities of interest of the databases considered is presented in Table 1.

The tissue segmentation of the two databases has a comparable accuracy, since the pipeline used in the eHead40 segmentation follows the standard steps used for the segmentation of MRI-derived “Virtual Family” models.

### 2.2. Electromagnetic Characterization and Simulations Settings

The electromagnetic commercial software Sim4Life V8.2 [57] was used to set up and perform simulations. This utilizes the quasi-static Laplace Equation (1):(1)∇(σ∇φ)=0
which is valid for the near-DC frequency range as for tDCS [37,39,46], to determine the electric potential distribution (φ) inside the human models due to stimulation. In Equation (1), *σ* (S/m) is the electrical conductivity of the tissues. The electric field distribution in each point of the conductive medium was obtained by means of Equation (2):(2)EF=−∇φ

The conductivities of the head tissues were assigned according to the data collected in the IT’IS low-frequency tissue properties database [61]. The electrical properties of the tissues mostly involved in tDCS [62] are *σ_skin_* = 0.148 S/m, *σ_fat_* = 0.078 S/m, *σ_CSF_* = 1.879 S/m, *σ_grey matter_* = 0.419 S/m, *σ_white matter_* = 0.348 S/m, *σ_bone cancellous_* = 0.08 S/m, *σ_bone cortical_* = 0.0063 S/m.

In all simulations, electrodes were modelled as rectangular pads (3 × 3 cm^2^ and 7 × 5 cm^2^ for anode and cathode, respectively) of 1 mm thick copper (σ = 5.9 × 10^7^ S/m) placed above a conductive sponge with the same dimensions (σ = 1.4 S/m [63]) and a thickness of 5 mm. Figure 1 shows the geometry of the problem: the anode placed over the posterior parietal cortex and the cathode over the contralateral supraorbital area. Specifically, the electrodes were positioned according to the international 10-10 EEG system in P2 (anode) and AF3 (cathode). According to the cranio-cortical correlation found in [64], P2 corresponds to the precuneus, a region implicated in abnormalities observed in subjects with social anxiety disorders.

The two electrodes were set to a fixed potential (+/−1V), and the results were later scaled to simulate a current flowing through the electrodes equal to 0.75 mA, corresponding to the fixed-dose tDCS approach used in Romero and colleagues’ study [24].

The computational domain was truncated at the level of the neck to limit the computational cost, and it had dimensions of 65 × 40 × 40 cm^3^. The domain was discretized through a non-uniform hexahedral mesh, with a mesh cell resolution ranging from 0.5 mm, for the brain tissues and the electrodes, to 2 mm elsewhere, to guarantee a fine discretization of tiny structures (such as the skin) and a reasonable computational cost. The resulting applied meshes ranged from about 90 to 130 million cells.

### 2.3. Analyzed Quantities and Statistical Analysis

For each simulation, on both databases, the following quantities were either extracted or calculated from the simulation results:*Anatomical quantities*: Cerebrospinal fluid volume, which due to its high conductivity shunts the injected current away from the target region; grey and white matter volumes (cm^3^); semi-circumference of the head (mm) in correspondence to the EEG 10-10 system points Fp and O, considered as a directly measurable anatomical characteristic of head size [65]; distance of P2–grey matter (mm) considered as a surrogate of the skull thickness (Table 1), one of the major determinants of current’s passage from the skin into the brain.*Electric field distribution*: Amplitude of the electric field distribution in grey and white matter.

For each subject, the 50th and 75th percentiles and the “peak” (i.e., the 99th percentile to filter spurious points due to staircase errors [40]; in the following it will be named “MaxEF”) of the electric field amplitude distribution over the white and grey matter were estimated. The spread of the field was calculated as the percentage volume of white and grey matter, where the EF amplitude was greater than the 50%, 70%, 80%, and 90% of a certain threshold defined on the responding subjects of the *Clinical Outcome Database* (for details see below).

To assess how morphological differences affect inter-subject variance in tDCS after-effects, possible correlations between anthropometric and EF quantities were quantified. They were determined by calculating the Pearson correlation coefficient r, under the condition that the variable distributions are normal and exhibit a monotonic trend, verified through the Shapiro–Wilk test and scatter plot, respectively. The Pearson correlation coefficient is a measure of the intensity of the linear correlation between the two variables. The closer to +/−1, the stronger the correlation is [66]. In simple linear regression, it coincides with the coefficient of determination (R-squared), so it expresses the proportion of total variance explained by the variable involved in the correlation investigated. A correlation is considered significant if the *p*-value is below 0.05.

The statistical analysis followed three main steps:Correlation between the anatomical quantities and the percentiles of the EF distribution across each individual’s brain from both databases (*Clinical Outcome* and *Integrated Databases*). By including all twenty-three simulations, the post hoc power statistics for the two main correlations investigated reached a power of 99% for a significance level alpha of 0.05. Additionally, correlations between anatomical quantities and age were also explored for all twenty-three subjects.Correlation between TEPs and the EF quantities across subjects only from the *Clinical Outcome Database* (twelve subjects): to assess which of the EF quantities analyzed—percentiles or spread—affects more the clinical outcome and then the subjects’ responsiveness.Correlation between the anatomical quantities and the EF quantity found in step 2 across the models of the *Clinical Outcome Database* (twelve subjects). Then a multiple regression model, solely on the models of the *Clinical Outcome Database*, was investigated [43] to assess a priori whether a subject will be respondent or not with the fixed dose of 0.03 mA/cm^2^.

The anatomical quantities considered for the regression models were CSF volume, skull thickness, head semi-circumference, and age. To avoid the multicollinearity of the independent variables, a pairwise linear correlation was performed, and variance inflation factors were computed. From a model which included all the aforementioned anatomical quantities, a process of backward elimination led to the development of two simplified models.

Several metrics were evaluated to assess and compare the quality of the regression models: root mean square error (RMSE), adjusted R-squared, Akaike information criterion (AIC) and Bayesian information criterion (BIC). RMSE gives a quantification of the accuracy of the model. The AIC, BIC, and adjusted R-squared consider not only the performance of the model but also its complexity: with a higher number of predictors, the model fits better data but introduces uncertainty into the estimation.

Since the regression coefficients are determined by minimizing the sum of squared errors, it was also checked, through the Kolmogorov–Smirnov test [67], that the residuals of the models followed a normal distribution.

Moreover, for both databases, the age was known, and the clinical outcome, in terms of early components of TEPs considered as a proxy of cortical excitability, was available for the *Clinical Outcome Database.*

## 3. Results

Figure 2 shows, as an example, the distribution of the amplitude of the electric field, normalized to an input current of 1 mA, on Duke’s grey matter surface. It shows how the stimulation mainly targets the anatomical regions under the two electrodes with an amplitude in close agreement with the literature studies [39,40,41,43,45,63,68,69,70].

*a*.
*Inter-variability of electric field distribution and anthropometric quantities*


The coefficient of variation, which is an index of the dispersion of the data, of the MaxEF and CSF volume over the whole database (i.e., joint *Clinical Outcome* and *Integrated Databases*) results in 36.5% and 30.5%, respectively. Leveraging on these results, which indicate a large variability in EF metrics and anatomical characteristics across subjects, this study explores a possible explanation for such a variability by investigating the correlation between anthropometric and electric quantities as reported in the following paragraphs.

*b*.
*Correlations between anthropomorphic quantities and electric field distribution percentiles*


Table 2 presents the Pearson correlation coefficients between anatomical quantities and the percentiles of the EF distributions (EF50, EF75, MaxEF). CSF volume in the brain region correlates negatively and significantly with EF50 and EF75; consequently by increasing the CSF volume, the amplitude of EF decreases. Conversely, the correlation with MaxEF is less robust. Other quantities such as white matter volume, distance of P2–grey matter (used as a surrogate for skull thickness), head semi-circumference, and age show a negative and significant correlation with all the percentiles of the EF distribution investigated, with *p*-values even lower than 0.0001.

Age is one of the demographic factors influencing EF distribution. Therefore, correlations between age and the other anthropomorphic quantities were explored. Our findings suggest that the negative correlation between age and MaxEF can be explained by the increase in both CSF volume (r = 0.43, *p* = 0.04) and skull thickness (r = 0.82, *p* < 0.001) with age. Children and adolescents have an average MaxEF value of 0.429 V/m, while adults (over 20 years old) exhibit a lower average peak value of 0.220 V/m. The analysis of variance, one-way ANOVA [71], showed that the difference between the two groups is highly significant (*p*-value < 0.0001).

*c*.
*Correlations between electric field quantities and TMS evoked potentials (TEPs) across Clinical Outcome Database*


The following investigation, conducted on the *Clinical Outcome Database* (twelve subjects), where TEP values were provided, aims to identify which EF quantity most correlates with increased cortical excitability.

Among the distribution percentiles and spread for EF, MaxEF and V50, respectively, show the strongest and most significant correlation with both TEP measurements.

Table 3 shows strong positive correlations between MaxEF and the cortical excitability. However, the analysis fails to identify a MaxEF value that distinguishes between responders and non-responders, and as was previously shown, MaxEF does not correlate significantly with the CSF volume, considered as a crucial anatomical quantity for dose personalization. Despite this, the strong correlation revealed the possibility of defining a reference value for responsiveness, calculated as the average of the MaxEF in the respondent subjects, which is 0.227 V/m.

The second EF quantity investigated, the spread of the EF, was computed as the percentage of brain tissues above different percentages of the established reference value. It gives an insight into the brain volume exposed to significant levels of EF levels during stimulation. Notably, V50 shows a strong correlation with cortical excitability (Table 3), and a threshold of V50 = 15.6%, found through a cluster-based approach, allows one to distinguish between responders and non-responders to tDCS treatment, with just one misclassified (S11), as shown in Figure 3. Moreover, V50 shows a high significant negative correlation with the amount of CSF (r = −0.9304, *p* < 0.0001).

The threshold of responsiveness (TEP = 1.1) [24], indicates at least a 10% increase in cortical excitability after a tDCS session compared to the baseline value.

*d*.
*Correlations between anatomical quantities and V50 (volume of brain tissues over 50% of the threshold 0.227 V/m) and multiple regression model*


Ultimately, this study sought to develop a multiple regression model capable of explaining V50 behaviour through anthropomorphic variables and predicting a subject’s responsiveness. This could provide valuable insights for calibrating doses to enhance individual responsiveness.

In the first (*Model 1*), the V50 is explained by the CSF volume (cm^3^) and skull thickness and has the form (3)(3)V50=a+b×CSF+c×skull thickness

The estimated coefficients are all significant and have the following values: *a =* 45.5%, *b =* −0.0653%/cm^3^, *c =* −0.746%/mm.

Since the *CODb* contains only twelve samples, the other model investigated was the simple linear regression model with CSF volume as sole variable (*Model 2*) (4):(4)V50=a+b×CSF

The model found has an adjusted R-square of 0.852. The estimated coefficients are *a* = 34.983%, *b* = −0.0736%/cm^3^, both with a *p*-value < 0.0001.

For both models, an analysis of variance and of the residuals were performed, yielding a variance of the residuals significantly lower than the one of V50, confirming that the included variables account for the entire V50 variance. Moreover, the residuals follow a normal distribution, as verified by the Kolmogorov–Smirnov test.

When the two models are compared, the model which includes both CSF volume and skull thickness performed better. It had a lower RMSE, AIC and BIC and higher adjusted R-squared, indicating a superior predictive performance.

From the two models it is possible to find thresholds of responsiveness in terms of anatomical quantities.

*Model 1* is shown in Figure 4. To assess a subject’s responsiveness, the combination of CSF and skull thickness must satisfy the condition that V50 is greater than 15.6%. In contrast, for *Model 2* the threshold is more straightforward: CSF = 263 cm^3^ with a 95% confidence interval of [243–284]cm^3^, as shown in Figure 5.

To test the possibility of generalizing the two models, a one-way ANOVA was performed on the MaxEFs. While the *IDb* over 20 years old is not significantly different from the *CODb*, the mean MaxEFs of the *IDb* under 20 are significantly different, with a very high F-score (209.62) and a *p*-value << 0.0001.

## 4. Discussion

tDCS represents a promising neuromodulation tool, but its use is still hampered by the high inter-subject variability in after-effects, largely owing to the lack of a framework for personalizing the injected current dose. The cortical electric field (EF) magnitudes found in this study align with those in the existing literature [40,70], with values around hundreds of mV/m (Figure 2).

The substantial variability in EFs found, with a coefficient of variation of 36.45% across subjects, when the electrode montage and the dose delivered in tDCS are fixed, highlights the weakness of the “one-size-fits-all” approach [38,72].

This study provides key insights into the primary determinants of the electric field distribution in relation to anatomical quantities (Table 2). Significant negative correlations were found between several anatomical characteristics and percentiles of the EF distribution in brain tissues, with both lower (EF50) and higher values (MaxEF). Specifically, the EF decreases with increasing cerebrospinal fluid (CSF) volumes due to the higher conductivity of CSF compared to the one of brain tissues, thus shunting the current away from the target brain area [69,73]. The magnitude of the EF decreases inversely with distance from the source and is influenced by the dielectric properties of tissues between the anode and brain, e.g., the skull’s low conductivity. Additionally, the semi-circumference of the head (HC) exhibits a strong negative correlation with EF and has been investigated as an alternative measure of CSF volume when an MRI scan is unavailable [65]. However, the significance of this correlation decreases when considering subjects of a similar age. Therefore, HC was not considered in the dose-tuning formula proposed in this study.

Correlations were conducted including all 23 head models to increase the posthoc power statistics for the two main correlations investigated (CSF-MaxEF, skull thickness–MaxEF).

Interestingly, the amount of brain volume exposed to significant levels of EF during stimulation (Table 3, Figure 3) was identified as the EF quantity most affecting subjects’ responsiveness (measured by TEPs). The responsiveness is more likely to occur when more than the 15.6% (the V50 threshold) of the brain is exposed to an EF higher than 0.114 V/m (i.e., 50% of 0.227 V/m, which is the mean of the peaks of the electric field for respondent subjects). This research is the first systematic effort that allows one to emphasize that the efficacy of tDCS depends more on the volume of brain tissues exposed to an EF above a certain threshold rather than solely the maximum electric field reached in brain tissues. This shift in perspective could prompt further investigations to validate these findings.

Two linear regression models were investigated to find a priori the responsiveness of a subject (Figure 4 and Figure 5). One model explained the V50 through both the CSF volume and the skull thickness, while the other relied only on the CSF volume. Although the simpler regression model has a worse adj-R2 and root mean squared error, it gives straightforward information with 95% confidence:CSF < 243 cm^3^: Subject is likely to respond.CSF > 284 cm^3^: Subject is likely to be non-respondent.

In this latter case, a higher dose of injected current is required to effectively increase cortical excitability, while for the former a dose of a current density of 0.03 mA/cm^2^ seems to give the desired result and, in perspective, could be decreased. For values of CSF inside the confidence of interval around 263 cm^3^, it is better to include in the prediction the skull thickness information to better discriminate between responders and non-responders.

The linear regression models developed in this study offer a practical tool for designing tDCS stimulation protocols that account for individual anthropometric variables. The V50 has been identified as a valuable surrogate for assessing subject responsiveness in terms of electric field spread, though further investigations are required to refine the exact relationship between the dose injected and the desired V50 and to validate the regression model’s accuracy and generalizability.

This study also investigated the impact of age on both anthropomorphic and electrical quantities, suggesting the necessity to analyze the responsiveness of children and adolescents separately from adults, as the developing brain exhibits different responses to the same stimulation [74]. Specifically, children tend to experience a higher and more focused EF than adults, which can be attributed to the positive correlation between age and both CSF volume [40] and skull thickness. These age-related correlations underscore the need to personalize dose calculation for children, as simple extrapolation from adult studies is inadequate. The computational study of Kessler and colleagues [69] reported how in electrotherapy, as in pharmacotherapy, dose selection in children requires special attention, and simple extrapolation from adult studies may be inadequate. This finding is consistent with the results obtained in this study, where the one-way ANOVA suggests that the mean of the EF peak in subjects under 20 years old significantly differs from subjects over 20 years old. Consequently, the simple extrapolation of an adult dose to paediatric cases may lead to suboptimal or unintended effects.

## 5. Conclusions

Computational and experimental studies have demonstrated the limitations of the “one-size-fits-all” approach in tDCS protocols [38,72], yet little has been done to address this issue. The literature lacks studies quantifying the relationship between anatomical factors and the electric field in targeted brain areas, and even fewer have proposed predictive models for individual responsiveness [75].

This study focuses on the main anatomical determinants of electric field distribution, providing insightful information into the relationship between clinical outcomes and electric field features. It measured the correlation between the electric field distribution in brain tissues and anthropometric factors and investigated the possibility of predicting subjects’ responsiveness through linear regression models, offering a practical tool for designing personalized tDCS protocols that account for anthropometric variables, such as CSF and skull thickness.

This study also investigates the necessity to analyze the responsiveness of children and adolescents separately from adults, since the developing brain exhibits different responses to the same stimulation.

This research is the first systematic effort that allows one to emphasize that the efficacy of tDCS depends more on the volume of brain tissue exposed to an electric field above a certain threshold (V50 = 15.6%, threshold = 0.227 V/m) rather than on the maximum electric field reached in a target region. This finding has important implications for the personalization of tDCS protocols, suggesting that dosing strategies should prioritize the extent of field exposure across the brain rather than peak values in localized areas.

Furthermore, this result raises several open questions, such as the following: *What is the optimal injected current to achieve the desired V50? Can we use the predictive model found to accurately estimate the V50 using only anthropometric measures, such as cerebrospinal fluid volume?* As our work provides only a preliminary analysis, further investigations are needed to address these questions.

A future development of the proposed work could be to improve CSF reconstruction by integrating T1- and T2-weighted images. Including axonal connectivity, obtained from diffusion tensor imaging (DTI), would provide deeper insights into the heterogeneity of tDCS after-effects.

Future efforts should concentrate on integrating the models and results of this study to design new tDCS protocols based on an individualized dosing tailored to specific individual anthropometric characteristics.

## Figures and Tables

**Figure 1 bioengineering-12-00656-f001:**
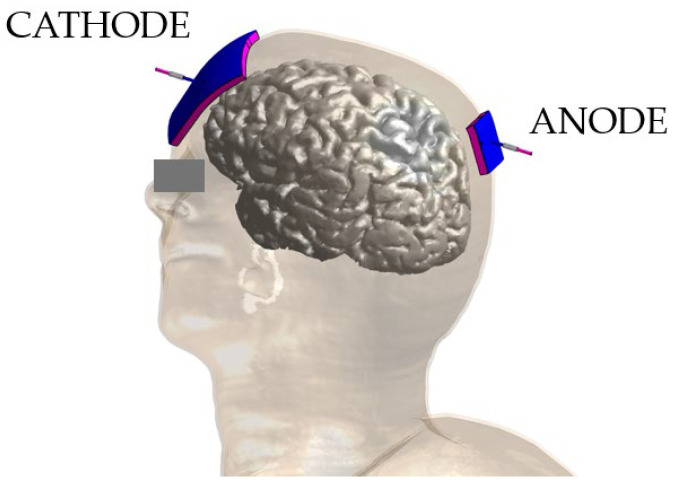
Duke model with the two electrodes placed in P2 and AF3.

**Figure 2 bioengineering-12-00656-f002:**
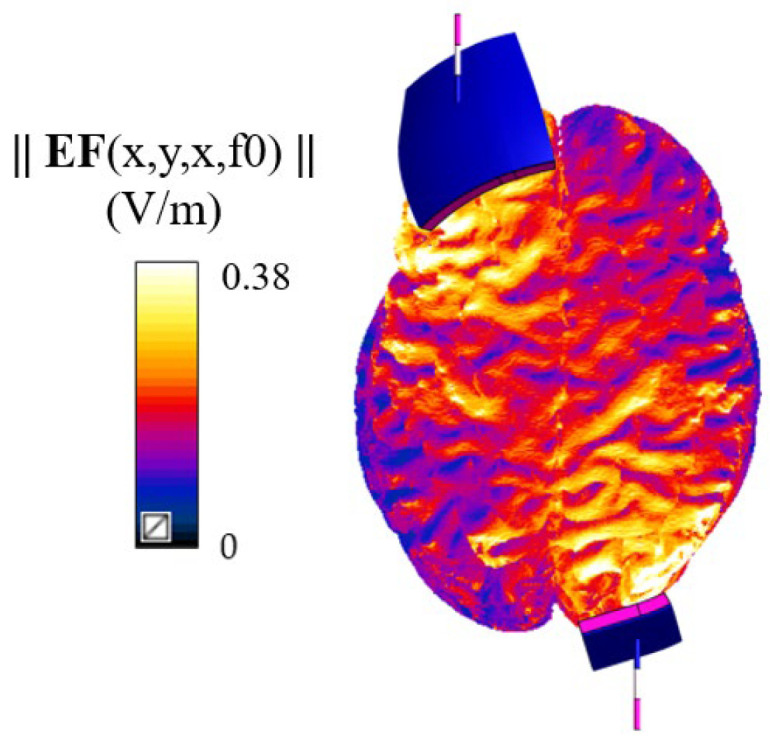
Distribution of the amplitude of the EF (V/m) on Duke’s grey matter surface.

**Figure 3 bioengineering-12-00656-f003:**
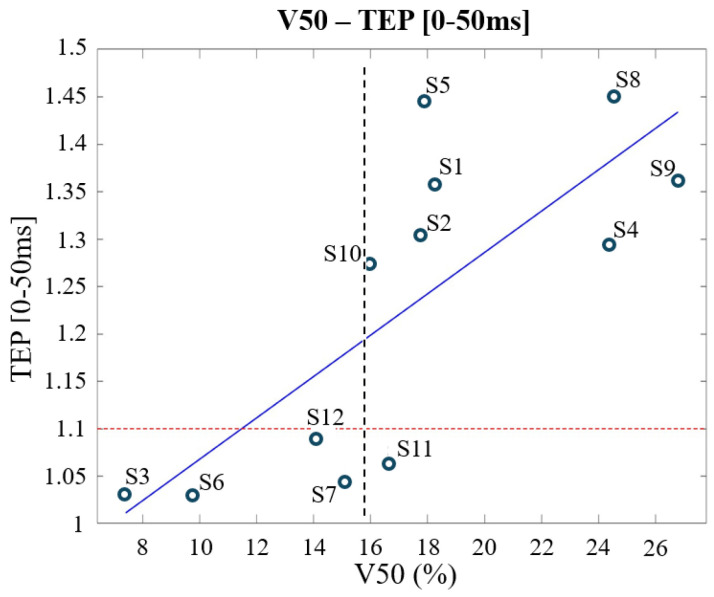
First TEP component (*y*-axis) behaviour with a varying V50 (*x*-axis, volume of brain tissues over the 50% of the threshold 0.227 V/m) across the *Clinical Outcome Database,* subjects are labeled as S1, S2, etc. The dotted red line represents the threshold of responsiveness (TEP = 1.1), while the dotted black line represents the identified threshold in terms of V50 (15.6%).

**Figure 4 bioengineering-12-00656-f004:**
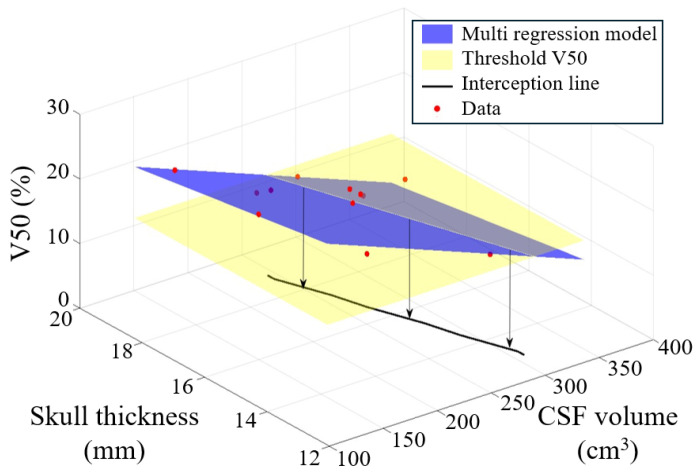
Mesh plot representing the multiple regression linear model (blue) and the threshold in terms of V50 (yellow). The black line shows the threshold of responsiveness in terms of CSF and skull.

**Figure 5 bioengineering-12-00656-f005:**
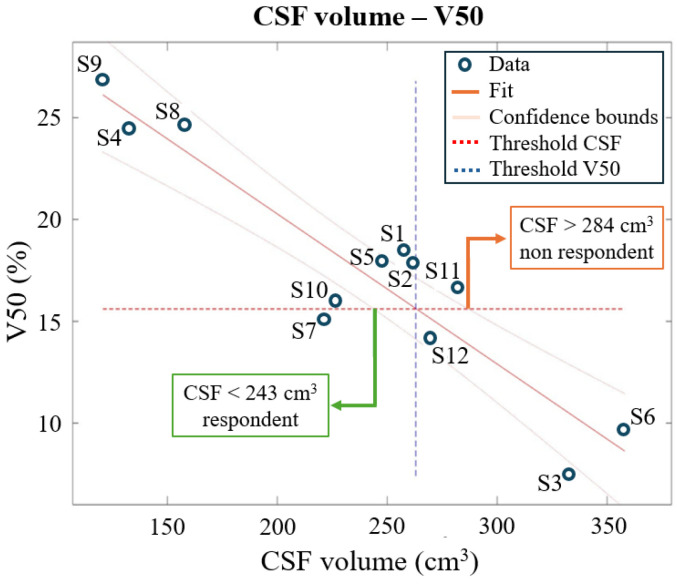
V50 (*y*-axis) behaviour with a varying CSF volume (*x*-axis) across the Clinical Outcome Database. The dotted red line represents the found threshold of responsiveness of V50 (V50 = 15.6%). The two thresholds in terms of CSF are outlined (green and orange boxes). They guarantee a 95% confidence. CSF and V50 have a very strong negative correlation, with r = −0.9304 and *p* < 0.0001.

**Table 1 bioengineering-12-00656-t001:** Mean and standard deviation of the main anatomical characteristics of the databases.

	Age (y)	Semi-Circumf Head (mm)	Skull Thickness (mm)	CSF Volume (cm^3^)	Grey Matter Volume (cm^3^)	White Matter Volume (cm^3^)
CODb ^1^	27 ± 6	277.22 ± 12.03	16.83 ± 2.17	238.97 ± 73.06	674.78 ± 116.61	562.9 ± 73.68
IDb over20 ^2^	34 ± 6	305.71 ± 41.96	18.83 ± 3.11	233.04 ± 35.78	598.82 ± 94.55	512.8 ± 81.08
IDb under20 ^2^	9 ± 3	258.29 ± 9.21	11.19 ± 1.73	198.48 ± 83.28	693.45 ± 49.49	370.67 ± 65.38

^1^ CODb = Clinical Outcome Database. ^2^ IDb over20/under20 = subjects from Integrated Database over/under 20 years.

**Table 2 bioengineering-12-00656-t002:** Correlation indexes between cerebrospinal fluid (CSF), white matter (WM) volumes, skull thickness (P2-GM), semi-circumference of the head, age, and 50th, 75th percentiles and the peak of the electric field distribution (EF50, EF75, MaxEF) calculated over the Clinical Outcome and Integrated Databases. Values marked with * are statistically significant (*p*-value < 0.05).

	EF50		EF75		Max EF	
	r	*p*	r	*p*	r	*p*
CSF volume	−0.82	0.0000 *	−0.791	0.0000 *	−0.4024	0.057
White matter volume	−0.6512	0.0008 *	−0.6605	0.0006 *	−0.7536	0.0000 *
Skull thickness	−0.5974	0.0026 *	−0.655	0.0007 *	−0.7918	0.0000 *
Semi-circumference	−0.4935	0.0196 *	−0.5447	0.0088 *	−0.6738	0.0006 *
Age	−0.5906	0.003 *	−0.6476	0.0008 *	−0.823	0.0000 *

**Table 3 bioengineering-12-00656-t003:** Correlation indexes between the two early components of TEPs and the peak of the electric field distribution (MaxEF) and the volume of brain tissues over 50% (V50) of the threshold 0.227 V/m calculated considering the Clinical Outcome Database. Values marked with * are statistically significant (*p*-value < 0.05).

TEP (ms)	Max EF		V50	
	r	*p*	r	*p*
0–50	0.7248	0.0077 *	0.7636	0.0039 *
50–100	0.5953	0.0412 *	0.5911	0.0429 *

## Data Availability

The data that support the findings of this study are available online at https://osf.io/fyx4h/?view_only=85ff1040331149878da23a0bf80a6048 (accessed on 12 June 2025).

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
