# Peer review of "Anatomical Characteristics Predict Response to Transcranial Direct Current Stimulation (tDCS): Development of a Computational Pipeline for Optimizing tDCS Protocols"

_bioengineering, 2025, doi:10.3390/bioengineering12060656_

Round 1
Reviewer 1 Report
Comments and Suggestions for Authors
My comments are below.
- The i-thenticate report is 18 %. Authors are encouraged to reduce it to below 15 %.
- What is the significance of (tDCS?
- Page 2, line 45 to 46 need to rewrite.
- Page 3, line 99 to 103 are not clear. Please revise it.
- Please elaborate P2 and AF3.
- Page 4, line 171, Anatomical quantities need to be explained.
- What is the limitation of one-size-fits-all?
- Please revise conclusion section.
- Check typos errors.
Reviewer 2 Report
Comments and Suggestions for Authors
The paper calls: “Anatomical characteristics predict response to transcranial direct current stimulation (tDCS) development of a computational pipeline for optimizing tDCS protocols” and concerned of analyze of transcranial direct current stimulation technique. The advantage of article actual topic of research and long references list (68 ones). However reviewer has some questions:
- May author can add typical electrical signal and more electro-technical details?
- What about race of people under research?
- Authors could add some unanswered question which were found under research.
Reviewer 3 Report
Comments and Suggestions for Authors
The authors' article focuses on computational modelling of transcranial direct current stimulation of the brain using detailed open MRI data for 23 patients of different sexes and ages. The present study was conducted with the objective of ascertaining the limitations of this non-invasive method for the treatment of neurological and psychological disorders. A study was conducted in order to investigate the effects of age and anatomical features, including cerebrospinal fluid volume, skull thickness, and brain tissue composition, on the distribution of electrical fields and cortical excitability during transcranial direct current stimulation. The findings indicated negative correlations between maximum electric field strength and anatomical variables.
The article is written in good English. It is very useful that the bibliography includes the Digital Object Identifier (doi). The most significant issue is that the authors paid disproportionately little attention to the primary results, which are outlined in Table 2. It is regrettable that they do not provide an explanation or a link to a source that would help readers understand what the Pearson correlation coefficient is. This section requires further elaboration.
The authors report that data from 23 patients were processed using mathematical methods. However, it is evident from Figures 3, 4 and 5 that the data points are limited to 12.
Please find below a number of comments concerning the text of the manuscript.
Due to the excessive use of abbreviations by the authors, the text becomes challenging to comprehend. There are too many abbreviations. Please, avoid using abbreviations in section headings and captions for figures and tables.
Lines 23-24 Change “Significant and negative correlations (p<0.05) were found in the data between the maximum electric field strength and anatomical variables.” By “Significant negative correlations (p<0.05) were found in the data between the maximum electric field strength and anatomical variable parameters.”
Table 1 is placed too far from its first citation.
Lines 142-145. This utilizes the quasi-static Laplace equation (1):
∇(𝜎∇𝜑) = 0 (1), which is valid ….
Line 213. give the reference for the Kolmogorov-Smirnov test
Line 247-248. It is not entirely clear where the given values came from, as they are not included in Table 2.
Line 250 Decipher ANOVA
Line 256. Among the distribution percentiles and EF spread for EF,
262-264 If you wouldn't mind, could you please rephrase this sentence? Correlation is not usually used, but it is revealed.
330-336 The magnitude of the EF decreases inversely with distance from the source and is influenced by the dielectric properties of tissues between the anode and brain, e.g. the skull's low conductivity. Additionally, the semi-circumference of the head (HC) exhibits a strong negative correlation with EF and has been investigated as an alternative measure of CSF volume when an MRI scan is unavailable [66]. However, the significance of this correlation decreases when considering subjects of a similar age. Therefore, HC was not considered in the dose-tuning formula proposed in this study.
Round 2
Reviewer 1 Report
Comments and Suggestions for Authors
Accepted in Present form
Reviewer 3 Report
Comments and Suggestions for Authors
The authors have modified the text of the article, taking all comments into account. If the editors deem the manuscript's content and scientific level appropriate for the journal's subject and scope, it can be accepted for publication.